# GLOBER: Coherent Non-autoregressive Video Generation via GLOBal Guided Video DecodER

**Mingzhen Sun** [1,2]     **Weining Wang** [1]     **Zihan Qin** [1,2]
**Jiahui Sun** [1,2]     **Sihan Chen** [1,2]     **Jing Liu** [1,2,*]
[1]Institute of Automation, Chinese Academy of Sciences (CASIA)
[2]School of Artificial Intelligence, University of Chinese Academy of Sciences (UCAS)
`sunmingzhen2020@ia.ac.cn` `weining.wang@nlpr.ia.ac.cn` `qinzihan2021@ia.ac.cn`
`sunjiahui19@mails.ucas.ac.cn` `sihan.chen@nlpr.ia.ac.cn` `jliu@nlpr.ia.ac.cn`

## Abstract

Video generation necessitates both global coherence and local realism. This work presents a novel non-autoregressive method GLOBER, which first generates global features to obtain comprehensive global guidance and then synthesizes video frames based on the global features to generate coherent videos. Specifically, we propose a video auto-encoder, where a video encoder encodes videos into global features, and a video decoder, built on a diffusion model, decodes the global features and synthesizes video frames in a non-autoregressive manner. To achieve maximum flexibility, our video decoder perceives temporal information through normalized frame indexes, which enables it to synthesize arbitrary sub video clips with predetermined starting and ending frame indexes. Moreover, a novel adversarial loss is introduced to improve the global coherence and local realism between the synthesized video frames. Finally, we employ a diffusion-based video generator to fit the global features outputted by the video encoder for video generation. Extensive experimental results demonstrate the effectiveness and efficiency of our proposed method[1], and new state-of-the-art results have been achieved on multiple benchmarks.

## 1 Introduction

When producing a real video, it is customary to establish the overall information (global guidance), such as scene layout or character actions and appearances, before filming the details (local characteristics) that draw up each video frame. The global guidance ensures a coherent storyline throughout the produced video, and the local characteristics provide the necessary details for each video frame. Similar to real video production, generative models for the video generation task must synthesize videos with coherent global storylines and realistic local characteristics. However, due to limited computational resources and the potential infinite number of video frames, how to achieve global coherence while maintaining local realism remains a significant challenge for video generation tasks.

Inspired by the remarkable performance of diffusion probabilistic models [1; 2; 3], researchers have developed a variety of diffusion-based methods for video generation. When generating multiple video frames, strategies of existing methods can be devided into two categories: autoregression and interpolation strategies. As illustrated in Fig. 1(a), the autoregression strategy [4; 5] first generates an initial video clip, and then employs the last few generated frames as conditions to synthesize subsequent video frames. The interpolation strategy [6; 7], depicted in in Fig. 1(b), generates keyframes first and then interpolates adjacent keyframes iteratively. Both strategies utilize generated video frames as conditions to guide the generation of subsequent video frames, enabling subsequent

---

[1]Our codes have been released in `https://github.com/iva-mzsun/GLOBER`

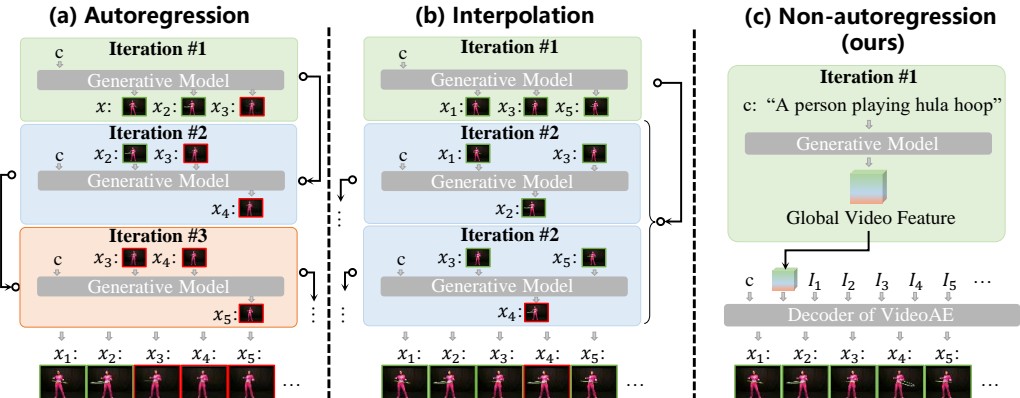

Figure 1: Three strategies for multi-frame video generation. (a) Autoregression strategy first generates the starting video frames and then autoregressively predicts subsequent video frames. This approach is prone to accumulating errors [6], such as the fading of the hula hoop. (b) Interpolation strategy generates video keyframes first and then iteratively interpolates adjacent keyframes. This approach can result in suboptimal video consistency since it is unaware of the global content. (c) Our proposed non-autoregression strategy first generates a global feature to provide global guidance, and then synthesizes local characteristics of video frames in a non-autoregressive manner. $c$ names the condition input of video description. $x_i$ and $I_i$ denote the $i$-th video frame and its index.

global storylines to be predicted from the context of the given frames. However, since the number of conditional video frames is limited by available computational resources and is generally small, these strategies have a relatively poor capacity to provide global guidance, resulting in suboptimal video consistency. In addition, the generation of local characteristics of subsequent frames refers to previous single frames, which can lead to error accumulation and distorted local characteristics [6].

In this paper, we present a novel non-autoregression method GLOBER, which first generates 2D global features to serve as global guidance, and then synthesizes local characteristics of video frames based on global features to obtain coherent video generation, as illustrated in Fig. 1(c). To be specific, we propose a video auto-encoder that involves a video encoder and a diffusion-based powerful video decoder. The video encoder encodes each input video into a 2D global feature. The video decoder decodes the storyline from the generated global feature and synthesizes necessary local characteristics for video frames. To achieve maximum flexibility, our video decoder is designed to involve no temporal modules and thus decode multiple video frames in a non-autoregressive manner. In particular, normalized frame indexes are integrated with generated global features to provide temporal information for the video decoder. In this way, we can obtain arbitrary sub video clips with predetermined starting and ending frame indexes. To leverage the success of image generation models, we initialize the video decoder with a pretrained image diffusion model [8]. Then the video auto-encoder can be trained in a self-supervised manner with the target of video reconstruction. Furthermore, we propose a Coherence and Realism Adversarial (CRA) loss to improve the global coherence and local realism in the decoded video frames. For video generation, another diffusion model is used as the video generator to generate global features by fitting the output of video encoder.

Our contributions are summarized as follows: (1) We propose a novel non-autoregression strategy for video generation, which provides global guidance while ensuring realistic local characteristics. (2) We introduce a powerful video decoder that allows for parallel and flexible synthesis of video frames. (3) We propose a novel CRA loss to improve global coherence and local realism. (4) Experiments show that our proposed method obtains new state-of-the-art results on multiple benchmarks.

## 2    Related Work

Current video generation models can be divided into three categories: transformer-based methods that model discretized video signals, generative adversarial networks (GAN) and diffusion probabilistic models (DPM). The first category encodes videos to discrete video tokens and then models these tokens with transformers [9; 10; 11; 12; 13; 14]. MOSO [12] decomposes video scenes, objects,

and their motions to cover video prediction, generation, and interpolation tasks in an autoregressive manner. TATS [13] follows the interpolation strategy, and introduces a time-agnostic VQGAN and a time-sensitive hierarchical transformer to capture long temporal dependencies

GAN-base methods [15; 16; 17] excel at generating videos in specific domains. MoCoGAN [17] proposes to decompose video content and motion by dividing the latent space. DIGAN [15] explores video encoding with implicit neural representations. StyleGAN-V [16] extends the image generation model StyleGAN [18] to the video generation task non-autoregressively. However, StyleGAN-V focuses on the division of content and motion noises and ignores the importance of global guidance. As a result, its content input (i.e., randomly sampled noise) contains limited information, whereas our global features can provide more valuable instructions.

DPM is a recently emerging method for vision generation [4; 19; 5; 20; 21]. VDM [4] is the first work that applies DPM to video generation by replacing all 2D convolutions with 3D convolutions. VIDM [22] generates videos in a frame-wise autoregression manner with two individual diffusion models. Both VDM and VIDM employ the autoregression strategy to obtain multiple video frames. In contrast, NUWA-XL [6] adopts the interpolation strategy but requires significant computational resources to support parallel inference. VideoFusion [19] targets at dividing the shared and residual video signals of different video frames and modeling them respectively. It employs a non-temporal diffusion model to synthesize video frames based on their indexes within the pixel space. PVDM [23] utilizes 3D-to-2D projection to encode a video into three 2D latent features for efficient video generation. In PVDM, the video decoder reconstructs video frames with fixed length (i.e. 16 frames) and fixed interval (i.e. predefined FPS). LVDM [24] encodes videos using 3D CNN through both spatial and temporal downsampling. Compared with previous methods, our method encompass the following advantages. Firstly, GLOBER can take advantage of the powerful generative capability of pretrained image diffusion models (e.g. stable diffusion) to synthesize reconstructed video frames, thus requiring a much smaller dimension of latent features to represent videos. Secondly, GLOBER is more flexible than prior works when decoding video frames from video latents. The video decoder in GLOBER can decode arbitrary video frames without length or interval limitations by taking the normalized indexes of target video frames as inputs. Thirdly, GLOBER is more efficient when training for video generation and synthesizing long videos. Fourthly, GLOBER obtains new state-of-the-art results on multiple benchmarks.

## 3    Method

In this section, we present our proposed method in details. The overall framework of our method is depicted in Fig. 2. A video sample is represented as $x \in R^{F \times H \times W \times C}$, where $F$ denotes the number of video frames, $H$ is the height, $W$ is the width, and $C$ is the number of channels. The $i$-th video frame is denoted as $x_i \in R^{H \times W \times C}$.

### 3.1    Video Auto-Encoder

The video auto-encoder comprises a video encoder and a video decoder. To reduce the computational complexity involved in modeling videos, an auxiliary image auto-encoder, which is known as KL-VAE [8] and has been validated in previous studies [8; 6], is employed to encode each video frame individually into a low-resolution feature. The video encoder takes video keyframes as input and encodes them into 2D global features. Then the video decoder is used to synthesize each video frame based on the corresponding frame index and the global feature.

**Frame Encoding**    VAEs [25; 26; 27; 8] are widely used models that reduce search space for vision generation. For high-resolution image generation, VAEs typically encode each image into a latent feature and then decode it back to the original input image [10; 27; 28; 29]. To reduce the spatial details of videos in a similar way, we employ a pretrained KL-VAE [8] to encode each video frame individually. Specifically, the KL-VAE downsamples each video frame $x_i$ by a factor of $f_{frame}$, obtaining a frame latent feature $z_i \in R^{H' \times W' \times C'}$, where $H'$ and $W'$ are $\frac{H}{f_{frame}}$ and $\frac{W}{f_{frame}}$, respectively, and $C'$ represents the number of feature channels.

**Video Encoding**    As illustrated in Fig. 2, the video encoder is composed of an embedding feature $e_v$, an input layer, a downsample module with a downsample factor of $f_{video}$, a mapping module, and

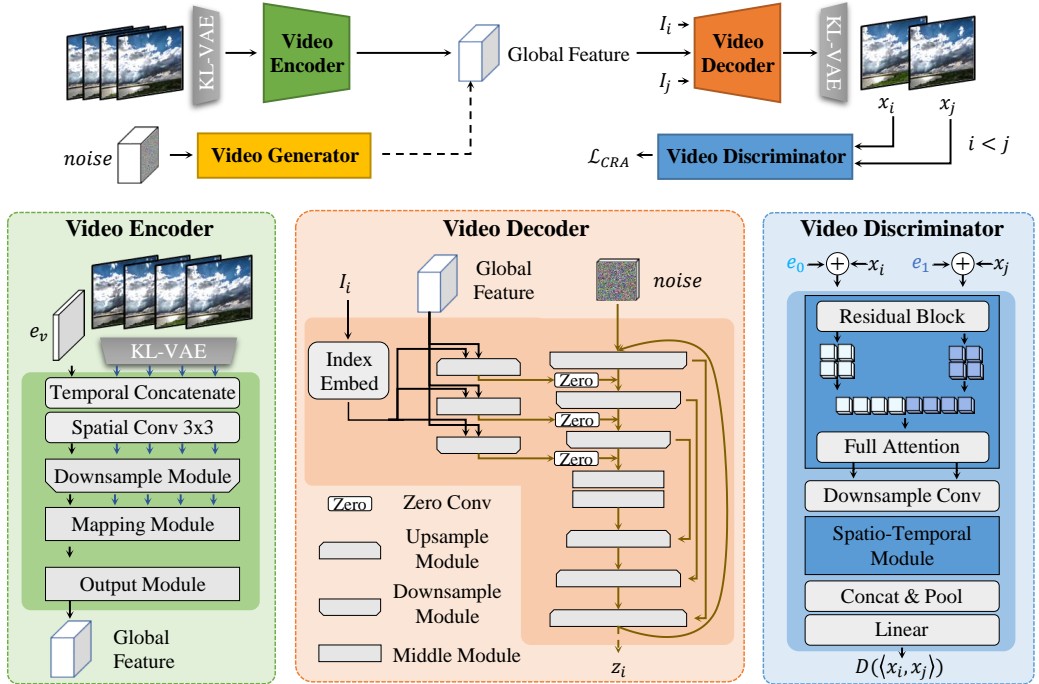

Figure 2: The overall framework of our proposed method GLOBER. During training, the video encoder and decoder are optimized jointly, with the video encoder encoding videos into global features, which are used by the video decoder to synthesize two randomly sampled video frames based on their corresponding frame indexes. We do not draw up the processing of timesteps and video descriptions in the video decoder for conciseness. The synthesized video frames are evaluated by the video discriminator for global coherence and local realism. Then the video generator is trained to generate global features by fitting the outputs of the video encoder. During generation, the video generator generates a novel global feature, which is then decoded by the video decoder to synthesize video frames in a non-autoregressive manner.

an output module. The embedding feature $e_v \in R^{H' \times W' \times C'}$ is randomly initialized and optimized jointly with the entire model. It should be noted that the temporal dimensions of the embedding feature and each frame feature are 1, which are omitted here for brevity. Since content redundancy exists between adjacent video frames [12], we select $K$ keyframes from each input video at equal intervals and encode them individually using KL-VAE. This process produces the corresponding frame features $z_{q_1}, z_{q_2}, ..., z_{q_K}$, where $q_k$ denotes the index of the $k$-th selected keyframe. Then, we concatenate the embedding feature $e_v$ with keyframe features along the temporal dimension, obtaining the input feature $[e_v : z_{q_1} : ... : z_{q_K}] \in R^{(K+1) \times H' \times W' \times C'}$ for the video encoder.

The input layer employs a simple 2D spatial convolution with a kernel size of 3 to expand the available channel maps from $C'$ to $D$, where $D$ represents the number of new channel maps. After the input layer, the downsample module processes these keyframe features to capture spatial and temporal information, while the mapping module follows. Specifically, the downsample module is composed of a residual layer, a spatial attention layer, a temporal attention layer, and a downsample convolution, while the mapping module follows a similar structure but replaces the downsample convolution with a temporal split operation. Finally, the output module comprises two spatial convolutions, a group normalization layer, and a SiLU activation function. It takes only the embedding part of the transformed feature as input and produces the mean and standard deviation (std) features of the global feature. Then the global feature can be sampled using the following equation:

$$v = v_{mean} + v_{std} * n \tag{1}$$

where $n$ is sampled from an isotropic Gauss distribution $\mathcal{N}(0, \mathbf{I})$, $v_{mean}, v_{std} \in R^{H'' \times W'' \times C''}$ are the mean and std features, $H''$ and $W''$ are $\frac{H'}{f_{video}}$ and $\frac{W'}{f_{video}}$ respectively, and $C''$ is the number of channels. Considering that we are going to model the global features for generation using a

diffusion-based model, which will be specified in Sec. 3.2, an additional kl loss [8] is employed to force the distribution of global features towards an isotropic Gauss distribution:

$$\mathcal{L}_{kl} = \frac{1}{2}(v_{mean}^2 + v_{std}^2 - \log v_{std}^2 - 1) \tag{2}$$

**Video Decoding**   We view the synthesis of video frames as a conditional diffusion-based generation task. As illustrated in Fig. 2, we employ UNet [8] as the backbone of the video decoder, which can be structured into the downsample, mapping, and upsample modules. Each module starts with a residual block and a spatial attention layer, while the downsample and upsample modules additionally include downsample and upsample convolutions respectively. Following [30], each frame feature $z_i^0$ (i.e. $z_i$) is corrupted by $T$ steps during the forward diffusion process using the transition kernel:

$$q(z_i^t|z_i^{t-1}) = \mathcal{N}(z_i^t; \sqrt{1-\beta_t}z_i^{t-1}, \beta_t\mathbf{I}) \tag{3}$$

$$q(z_i^t|z_i^0) = \mathcal{N}(z_i^t; \sqrt{\bar{\alpha}_t}z_i^0, (1-\bar{\alpha}_t)\mathbf{I}) \tag{4}$$

where $\{\beta_t \in (0,1)\}_{t=1}^T$ is a set of hyper-parameters, $\alpha_t = 1 - \beta_t$ and $\bar{\alpha}_t = \prod_{i=1}^t \alpha_i$. Based on Eq. (4), we can obtain the corrupted feature $z_i^t$ directly given the timestep $t$ as follows:

$$z_i^t = \sqrt{\bar{\alpha}_t}z_i^0 + (1-\bar{\alpha}_t)n_t \tag{5}$$

where $n_t$ is a noise feature sampled from an isotropic Gauss distribution $\mathcal{N}(0,\mathbf{I})$. The reverse diffusion process $q(z_i^{t-1}|z_i^t, z_i^0)$ has a traceable distribution:

$$q(z_i^{t-1}|z_i^t, z_i^0) = \mathcal{N}(z_i^{t-1}|\tilde{\mu}_t(z_i^t, z_i^0), \tilde{\beta}_t\mathbf{I}) \tag{6}$$

where $\tilde{\mu}_t(z_i^t, z_i^0) = \frac{1}{\sqrt{\alpha_t}}(z_i^t - \frac{\beta_t}{\sqrt{1-\bar{\alpha}_t}}n_t)$, $n_t \sim \mathcal{N}(0,\mathbf{I})$, and $\tilde{\beta}_t = \frac{1-\bar{\alpha}_{t-1}}{1-\bar{\alpha}_t}\beta_t$.

Given that the added noise $n_t$ in the $t$-th step is the only unknown term in Eq. (6), we train the UNet to predict $n_t$ from $z_i^t$ condition on the timestep $t$, the global feature $v$, the frame index $i$, and the video description $c$ if exists. To achieve this, a timestep and an index embedding layers, additional modules, and a text encoder are employed to process these condition inputs. In particular, the timestep embedding layer first obtain the sinusoidal position encoding [31] of the diffusion timestep $t$ and then passes it through two linear functions with a SiLU activation interposed between them. Following previous works [30; 32], the timestep embedding is integrated with intermediate features by the residual block in each UNet module. The index embedding layer embeds the frame index in a similar way with the timestep embedding layer. The additional modules (i.e. downsample, mapping, and upsample modules) are utilized to incorporate the index embedding with the global feature and to extract multi-scale representations of the corresponding video frame. These representations are then added to the outputs of UNet modules after zero-convolution. When encoding video descriptions into text embeddings, a pretrained model, namely CLIP [33], is utilized as the text encoder. During attention calculation, these text embeddings are concatenated with flattened frame features to provide cross-modal instructions. Finally, the L2 distance between the predicted noise $n(z_i^t, t, i, v, c)$ and the added noise $n_t$ is calculated as the training target:

$$\mathcal{L}_{rec} = \|n(z_i^t, t, i, v, c) - n_t\|_2 \tag{7}$$

where $\|*\|_2$ denotes the calculation of the L2 distance. The total training loss is:

$$\mathcal{L} = \mathcal{L}_{rec} + \lambda_1\mathcal{L}_{kl} + \lambda_2\mathcal{L}_{cra}^G \tag{8}$$

where $\lambda_1$ and $\lambda_2$ are hyper-parameters and $\mathcal{L}_{cra}^G$ is specified in the following paragraph. During generation, the video decoder synthesizes video frames given content features, frame indexes and video descriptions (if exist) by denoising noise features with multiple steps as in [30; 34].

**Coherence and Realism Adversarial Loss**   We propose a novel Coherence and Realism Adversarial (CRA) loss to improve global coherence and local realism of synthesized video frames. To reduce computation complexity, we randomly synthesize two video frames with indexes $i$ and $j$, where $0 \le i < j \le F$, using the video decoder. The synthesized video frame $\bar{x}_i$ can be decoded by KL-VAE from the frame feature $\bar{z}_i$, which is obtained directly in each training step with the following formulation:

$$\bar{z}_i^0 = \frac{1}{\sqrt{\bar{\alpha}_t}}z_i^t - \frac{(1-\bar{\alpha}_t)}{\sqrt{\bar{\alpha}_t}}n(z_i^t, t, i, v, c) \tag{9}$$

where $n(z_i^t, t, i, v, c)$ is the predicted noise of $n_t$. Then the CRA loss is calculated given the synthesized and real video frames using a video discriminator as depicted in Fig. 2.

In our case, we expect the discriminator to provide supervision on global coherence and local realism of synthesized video frames. To ensure the effectiveness of the CRA loss, we formulate it based on the following two guiding principles: (1) the real samples selected for the discriminator must exhibit the desired correlations, whereas the fake samples must deviate from the expected patterns and are often challenging to distinguish; (2) the real samples selected for the video auto-encoder (i.e. the generator) should serve as the fake samples for the discriminator for adversarial training. In particular, to make the discriminator aware of local realism, we select $\langle x_i, x_j \rangle$ as the real sample and $\langle x_i, \bar{x}_j \rangle, \langle \bar{x}_i, x_j \rangle, \langle \bar{x}_i, \bar{x}_j \rangle$ as fake samples according to the first principle. This is because the latter samples contain at least one synthesized video frame and violate the target pattern of realism. Furthermore, to ensure that the discriminator is aware of the global coherence, we utilize samples that violate temporal relationships like $\langle x_j, x_i \rangle$ and $\langle \bar{x}_j, \bar{x}_i \rangle$ as fake samples of the discriminator. Then, according to the second principle, $\{\langle \bar{x}_i, \bar{x}_j \rangle, \langle x_i, \bar{x}_j \rangle, \langle \bar{x}_i, x_j \rangle\}$ are chosen as real samples of the video auto-encoder, since these samples contain at least one synthesized video frames. The CRA loss can be formulated as $\mathcal{L}_{cra}^D$ for the discriminator and $\mathcal{L}_{cra}^G$ for the generator:

$$
\begin{aligned}
\mathcal{L}_{cra}^D =& log(1 - \mathcal{D}(\langle x_i, x_j \rangle)) + log\mathcal{D}(\langle x_i, \bar{x}_j \rangle) + log\mathcal{D}(\langle \bar{x}_i, x_j \rangle) \\
&+ log\mathcal{D}(\langle \bar{x}_i, \bar{x}_j \rangle) + log\mathcal{D}(\langle x_j, x_i \rangle) + log\mathcal{D}(\langle \bar{x}_j, \bar{x}_i \rangle) \\
\mathcal{L}_{cra}^G =& log(1 - \mathcal{D}(\langle \bar{x}_i, \bar{x}_j \rangle)) + log(1 - \mathcal{D}(\langle \bar{x}_i, x_j \rangle)) + log(1 - \mathcal{D}(\langle x_i, \bar{x}_j \rangle))
\end{aligned}
\tag{10}
$$

As illustrated in Fig. 2, the discriminator is built on several spatio-temporal modules that consist of residual blocks and spatio-temporal full attentions. Considering that traditional attention is invariant to the order of input features, two position embeddings $e_0$ and $e_1$ are employed and added to the previous video frame $x_i$ and the subsequent video frame $x_j$ respectively with $0 \le i < j \le F$. These position embeddings are randomly initialized and optimized at the same time with the discriminator.

## 3.2 Generative Model

As shown in Fig. 2, we can obtain any desired video clip by feeding the frame indexes, global feature, and corresponding video description (if available) to the video decoder. Since the global feature is the only unknown term, we can train a conditional generative model to generate a global feature based on the video description. Considering that global feature are 2D features, the generative model can be relieved from the burden of modeling intricate video details.

In practice, global features typically have a small spatial resolution, and we utilize a transformer-based diffusion model, DiT [35], to generate them. Specificcally, each 2D global feature $v \in R^{H'' \times W'' \times C}$ is flattened to a 1D feature with length $H'' \times W''$. Then the diffusion and reverse diffusion processes of $v_l$ are similar to the procedures outlined in Eq. (3) and Eq. (6), except that the only condition is the video description (if exists) in the training and generation processes.

# 4 Experiments

In this section, we first introduce the experimental setups in Sec. 4.1. Following this, in Sec. 4.2 and Sec. 4.3, we compare the quantitative and qualitative performance of our method and prior methods on four challenging benchmarks: Sky Time-lapse [36], TaiChi-HD [37], UCF-101 [38], and Webvid-10M [39]. Finally, Sec. 4.4 presents the results of ablation studies conducted to analyze the necessity of the CRA loss, and Sec. 4.5 explores the influence of global feature shape on the generation performance.

## 4.1 Experimental Setups

All experiments are implemented using PyTorch [40] and conducted on 8 NVIDIA A100 GPUs, with 16-precision adopted for fast training. During the training of the video auto-encoder, pretrained KL-VAEs [8] were utilized to encode each video frame $x_i$ into a latent feature $z_i$, with a downsample factor of $f_{frame} = 8$ for $256^2$ resolution and $f_{frame} = 4$ for $128^2$ resolution. The latent features were then of resolution $32^2$. The video encoder subsequently encoded the latent features of video keyframes, extracted at fixed frame indexes of $[0, 5, 10, 15]$ for 16-frame videos, with a downsample

Table 1: Quantitative comparison against prior methods on UCF-101, Sky Time-lapse and TaiChi-HD datasets for video generation. $c$ denotes the conditon of video descriptions.

(a) Sky Time-lapse w/o c

| Methods | FVD↓ |
|---|---|
| Resolution $256^2$ | |
| MoCoGAN-HD [41] | 164.1 |
| VideoGPT [42] | 222.7 |
| DIGAN [15] | 83.1 |
| StyleGAN-V [16] | 79.5 |
| GLOBER (ours) | **78.1** |

(b) TaiChi-HD w/o c

| Methods | FVD |
|---|---|
| Resolution $128^2$ | |
| SytleGAN-V [16] | 143.5 |
| DIGAN [15] | 128.1 |
| GLOBER (ours) | **124.2** |

(c) Webvid-10M w/ c

| Methods | FVD | CLIP |
|---|---|---|
| Resolution $256^2$ | | |
| VideoCrafters [2] | 759.3 | 0.2981 |
| LVDM [24] | 455.5 | 0.2751 |
| ModelScope [1] | 414.1 | 0.3000 |
| VideoFactory [43] | 292.4 | **0.3070** |
| GLOBER (ours) | **234.8** | 0.2816 |

(d) UCF-101 w/o c

| Methods | FVD |
|---|---|
| Resolution $128^2$ | |
| DIGAN [15] | 577 |
| TATS [13] | 420 |
| VIDM [22] | 263 |
| GLOBER (ours) | **239.5** |

(e) UCF-101 w/ c

| Methods | FVD |
|---|---|
| Resolution $128^2$ | |
| TGANv2 [44] | 1209 |
| DIGAN [15] | 465 |
| TATS [13] | 332 |
| MMVG [45] | 328 |
| CogVideo [9] | 305 |
| VIDM [22] | 263 |
| VideoFusion [19] | 173 |
| GLOBER (ours) | **151.5** |
| Resolution $256^2$ | |
| DIGAN [15] | 471.9 |
| Make-A-Video [46] | 367.2 |
| GLOBER (ours) | **168.9** |

Table 2: Quantitative comparison against prior methods on UCF-101 for unconditional video generation with $256^2$ resolution.

| Method | CogVideo [9] | MagicVideo [47] | LVDM [24] | ModelScope [1] | VideoLDM [7] |
|---|---|---|---|---|---|
| Zero-Shot | ✓ | ✓ | ✓ | ✓ | ✓ |
| FVD | 701.6 | 699.0 | 641.8 | 639.9 | 550.6 |
| Method | VideoCrafters [2] | VideoFactory [43] | VideoGPT [42] | MoCoGAN [17] | StyleGAN-V [16] |
| Zero-Shot | ✓ | ✓ | ✗ | ✗ | ✗ |
| FVD | 516.2 | 410.0 | 2880.6 | 2886.8 | 1431.0 |
| Method | CogVideo [9] | LVDM [24] | PVDM [23] | GLOBER (ours) | |
| Zero-Shot | ✗ | ✗ | ✗ | ✗ | |
| FVD | 626 | 372 | 343.6 | **252.7** | |

factor of $f_{video} = 2$ and a number of output channels of $C'' = 16$, resulting in global features of shape $16 \times 16 \times 16$. The dimension of $C'$ is 4 for 256x256 resolution and 3 for 128x128 resolution. We first train the video auto-encoder as well as the discriminator jointly until convergence, and then train DiT with the parameters of the video auto-encoder being fixed. The video auto-encoder was trained with a batch size of 40 per GPU for 80K, 40K, and 40K steps on the UCF101, TaiChi-HD, and Sky Time-lapse datasets, respectively. The loss weight $\lambda_1$ and $\lambda_2$ are set as 1e-6 and 0.1, respectively. When training the video generator, a Transformer-based diffusion model, DiT [35], was used as the backbone. The batch size was set to 32 per GPU, and the number of training iterations was 200K, 100K, and 100K for the UCF101, TaiChi-HD, and Sky Time-lapse datasets, respectively. When generating videos, we sample each global feature with the number of DDIM steps being 50 and the unconditional guidance scale being 9.0 expect otherwise specified. Then the video decoder synthesizes target video frames parallelly with the number of DDIM steps being 50 and the unconditional guidance scale being 3.0 for the Sky Time-lapse and TaiChi-HD datasets and 6.0 for the UCF101 dataset.

## 4.2 Quantitative Comparison

**Generation Quality** Table 1 and Table 2 reports the results of our model trained on the Sky Time-lapse, TaiChi-HD, UCF-101, and Webvid-10M datasets for 16-frame video generation in both unconditional and conditional settings. As shown in Table 1(a), Table 1(b), and Table 1(c), our method achieves comparable performance with prior state-of-the-art models on the Sky Time-lapse, TaiChi-HD, and Webvid-10M datasets, respectively. To comprehensively compare the performance

---

[1]https://github.com/modelscope/modelscope
[2]https://github.com/AILab-CVC/VideoCrafter

Table 3: Comparison of sampling time/memory using different methods for generating multiple video frames with resolution of $256^2$, batch size of 1, diffusion steps of default settings, and comparable GPU memory on a v100 GPU. $F$ represents the number of video frames. AR denotes autoregression, IP denotes interpolation and NAR denotes non-autoregression.

| Method | VIDM [22] | VDM [4] | LVDM [24] | PVDM [23] | VideoFusion [19] | TATS [13] | ModelScope | GLOBER (ours) |
|---|---|---|---|---|---|---|---|---|
| NAR | ✗ | ✗ | ✗ | ✗ | ✓ | ✗ | ✓ | ✓ |
| Video Encoding | ✗ | ✗ | ✓ | ✓ | ✗ | ✓ | ✓ | ✓ |
| $\text{NUM}_{F=16}$ | - | - | 12288 | 8192 | - | **4096** | 16384 | **4096** |
| $\text{NUM}_{F=128}$ | - | - | 98304 | 65536 | - | 32768 | 131072 | **4096** |
| Diffusion Steps | 100 | 100 | 50 | - | 50 | - | 50 | 50+50 |
| $F = 16$ | 192s/20G | 125s/11G | 75s/9G | - | 22s/7G | **6s**/16G | 31s/6G | **6s**/7G |
| $F = 32$ | 375s/20G | 234s/11G | 141s/13G | - | 39s/9G | 26s/16G | 48s/8G | **11s**/11G |
| $F = 64$ | 771s/20G | 329s/11G | 288s/20G | - | 76s/13G | 65s/16G | 82s/12G | **21s**/19G |

on the more challenging UCF-101 dataset, we conduct experiments on both unconditional and conditional video generation at resolutions of $128^2$ and $256^2$. For unconditional video generation, we use the fixed textual prompt "A video of a person" as the video description, while for conditional video generation, we use the class name of each video as the video description. Our proposed method significantly outperforms previous state-of-the-art models, as shown in Table 1(d), Table 1(e), and Table 2. The superior performance of our method can be attributed to two factors. Firstly, we explicitly separate the generation of global guidance from the synthesis of frame-wise local characteristics. As global features are 2D features with relatively small spatial resolution, our generative model can model them effectively and efficiently without imposing a high computational burden. Secondly, the synthesis of frame details can refer to the global feature, which provides global guidance such as scene layout, object appearance, and overall behaviours, making it easier for our method to achieve global-coherent and local-realistic video generation.

**Generation Speed**  We quantitatively compare the generation speed among various video generation methods and report the results in Table 3. Our GLOBER method demonstrates remarkable efficiency in generating video frames, thanks to its utilization of the non-autoregression strategy. In contrast, prior methods such as VIDM and VDM, which follow the auto-regression strategy, maintain unchanged memory but require significantly more time for generation. VideoFusion also adopts a non-autoregressive strategy for generating multiple video frames. However, VideoFusion remains slower than our GLOBER, since VideoFusion directly models frame pixels, which brings huge computational burden when generating multiple video frames. TATS employs the interpolation strategy by first generating video keyframes and then interpolating 3 frames between adjacent keyframes. However, it involves autoregressive generationg of video keyframes and the interpolation process is not parallelly implemented, thus being slower than our GLOBER. Notably, GPU memory of VideoFusion and our GLOBER increases with the number of video frames due to the parallel synthesis of all video frames.

### 4.3 Qualitative Comparison

As depicted in Figure 3, we conduct a qualitative comparison of our method with previous approaches on the UCF-101, Sky Time-lapse, and TaiChi-HD datasets. Samples of prior methods are obtained from [22; 19]. The UCF-101 dataset records 101 human actions and is the most challenging and diverse dataset. When applied to this dataset, GAN-based methods such as DIGAN and StyleGAN-V generate video samples that lack distinctiveness. In contrast, TATS, which is built on Transformer and utilizes the interpolation strategy, generates video samples that are more identifiable. In comparison to TATS, diffusion-based methods like VideoFusion [19] and VIDM [22] produce samples with more pronounced appearances. However, VideoFusion generates slightly blurred object appearances, and VIDM generates overly trivial object motions. Conversely, our GLOBER generates samples that exhibit both distinct appearances and conspicuous movements. On the Sky Time-lapse dataset, samples generated by DIGAN, StyleGAN-V, and TATS display trivial motions and simplistic objects. VideoFusion and VIDM generate samples with enhanced details and more discernible boundaries, while their motion remains somewhat negligible. In contrast, our GLOBER generates video samples with more dynamic movements and significantly richer visual details. Similarly, on the TaiChi-HD dataset, the human appearances generated by DIGAN, StyleGAN-V, and TATS are noticeably

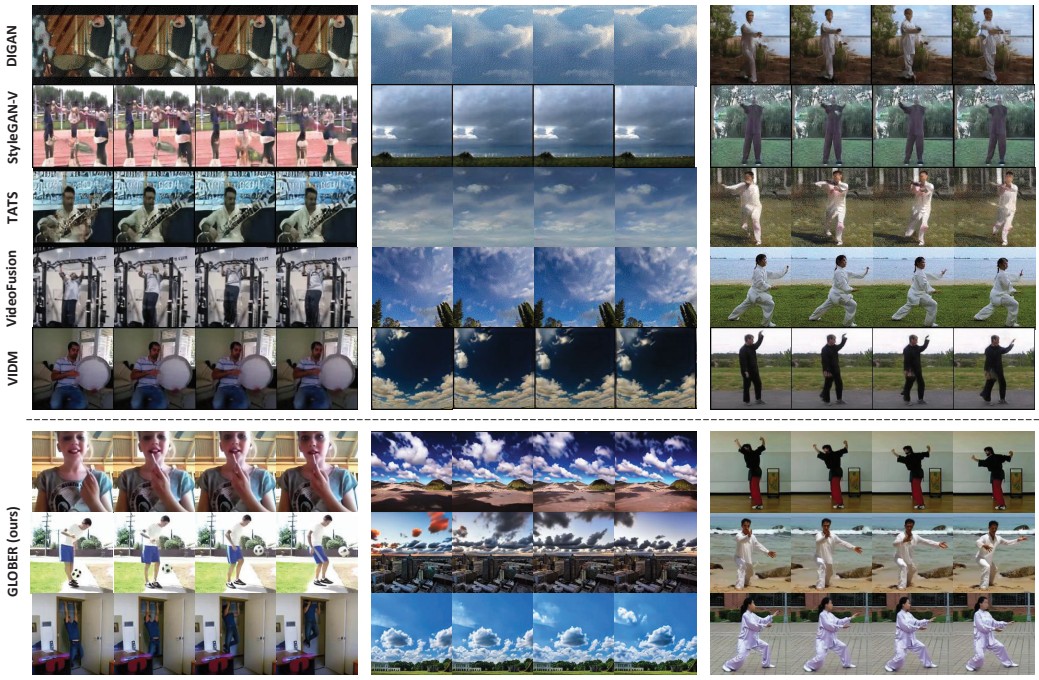

Figure 3: Quantitative comparison against previous methods on the UCF-101 (left), Sky Time-lapse (middle) and TaiChi-HD (right) datasets. Our showcased samples on the UCF-101 datasets are produced using the video descriptions "Apply Lipstick", "Soccer Juggling", and "Pull Ups", respectively. Samples on the Sky Time-lapse and TaiChi-HD datasets are generated using fixed video descriptions being "A time-lapse video of sky" and "Tai chi", respectively.

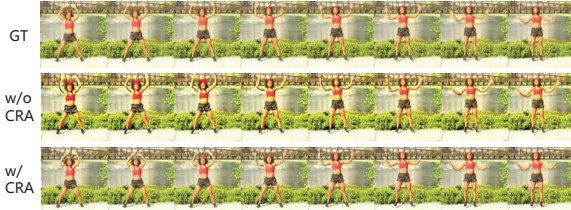

Figure 4: Visualization of samples synthesized with or without the CRA loss by the video auto-encoder.

Table 4: Ablation study on the CRA loss.

| Datasets | CRA loss | rFVD |
|---|---|---|
| UCF-101 | w/o | 106.7 |
| | w/ | **69.7** |
| Sky Time-lapse | w/o | 84.3 |
| | w/ | **63.5** |
| TaiChi-HD | w/o | 91.3 |
| | w/ | **60.1** |

distorted. While VideoFusion and VIDM achieve improved human appearances, their movements remain trivial. In contrast, samples generated by our GLOBER exhibit significant movements and distinct object appearances.

## 4.4 Ablation Study on the CRA Loss

To investigate the effectiveness of our proposed CRA loss, we conducted an ablation study where the CRA loss was removed during the training of the video auto-encoder. The quantitative results are presented in Table 4, which unequivocally demonstrate the effectiveness of our CRA loss on all three benchmarks: UCF-101, Sky Time-lapse, and TaiChi-HD, with a remarkable reduction in FVD scores by 37.0, 20.8, and 31.2, respectively. In addition, we qualitatively compare videos synthesized by models with or without the CRA loss in Fig. 4, which clearly shows that videos generated using the CRA loss exhibit better video consistency and more distinct local details, demonstrating the necessity of our proposed CRA loss. The effectiveness of our CRA loss can be attributed to two key factors. Firstly, by penalizing samples that violate temporal relationships, the CRA loss enhances the overall coherence of the synthesized video frames. Secondly, by utilizing pairwise video frames as inputs for

the discriminator, the CRA loss can identify and penalize video frames that exhibit inconsistent or distorted local characteristics, which further enhances the realism of the synthesized videos.

### 4.5 Impact on the Global Feature Shapes

We experiment the effect of different shapes of the global features on the quality of synthesized video frames. As reported in Table 5, videos generated using global features with a shape of $8 \times 8 \times 64$ exhibit lower quality than those generated using global features with a shape of $16 \times 16 \times 16$. And the use of global features with a shape of $16 \times 16 \times 32$ brings a further improvement of 39.2 in FVD score. It may be

Table 5: Sensitivity analysis on the shape of global features on the UCF-101 datasets.

| $H'' \times W'' \times C''$ | $\text{FVD}_{rec} \downarrow$ | $\text{FVD}_{gen} \downarrow$ |
|---|---|---|
| $8 \times 8 \times 64$ | 276.1 | 725.1 |
| $16 \times 16 \times 16$ | 189.7 | 560.4 |
| $16 \times 16 \times 32$ | 184.9 | 521.2 |

due to two reasons. Firstly, our experiments are conducted on the UCF-101 dataset, which contains complex motions that require more channels to be effectively represented. Secondly, by utilizing the KL loss to constrain the distribution of global features toward an isotropic Gaussian, it becomes easier for the generative model to fit the distribution of the global features even with a number of channels, thus increasing the number of channels could bring improved performance.

## 5 Conclusion and Limitations

This paper introduces GLOBER, a novel diffusion-based video generation method that emphasizes the significance of global guidance in multi-frame video generation. Our method offers three distinct advantages. Firstly, it alleviates the computation burden of modeling video generation by replacing 3D video signals with 2D global features. Secondly, it utilizes a non-autoregressive generation strategy, enabling the efficient synthesis of multiple video frames and surpassing the performance of prior methods in terms of efficiency. Lastly, by incorporating global features as guidance, our method generates videos with enhanced coherence and realism, achieving new state-of-the-art results on multiple benchmarks. Nevertheless, our research exhibits several limitations. Firstly, we find GLOBER difficult to process videos with frequent scene changes, as such transitions have the potential to disrupt the video coherence. Furthermore, we have not explored the performance of GLOBER in open-domain video generation tasks due to computational resource constraints. Future works are encouraged to solve the above issues.

## 6 Acknowledgements

This work was supported by the National Key Research and Development Program of China (No.2022ZD0118801), National NaturalScience Foundation of China (U21B2043,62102416, 62206279, 62102419).

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

# A    Appendix

# B    Broader Impact

The goal of this work is to advance research on video generation methods. Our method has the potential to facilitate the workflow of film production and animation, exhibiting a positive influence on creative video applications. Since our method is trained mainly on domain-specific datasets, the potential deleterious consequences of exploiting our model for malicious purposes, such as spreading misinformation or producing fake videos, seem to be insignificant. Nevertheless, it remains crucial to apply an abundance of caution and implement strict and secure regulations.

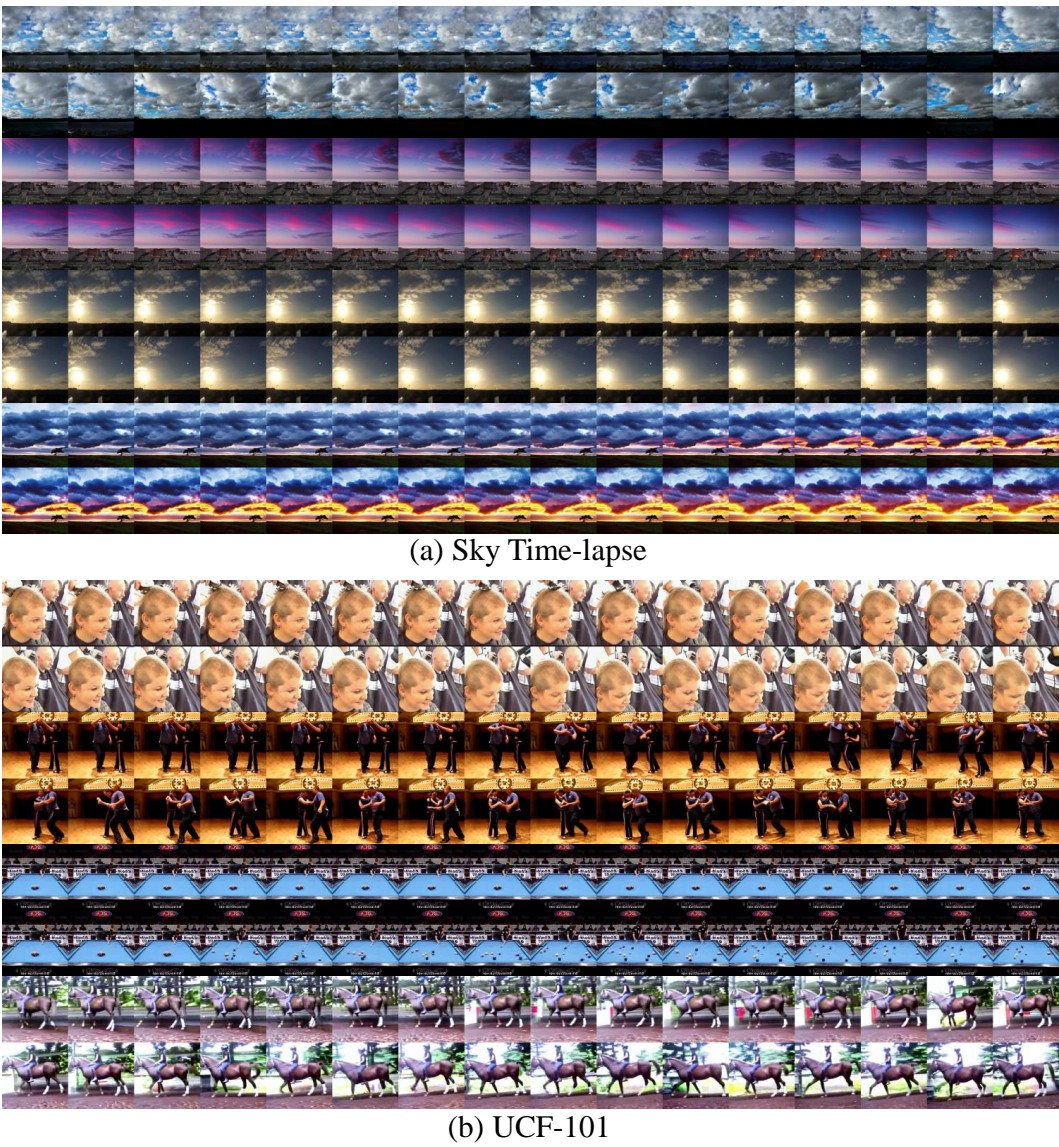

(a) Sky Time-lapse

(b) UCF-101

Figure 5: Genetated long videos with 128 frames on the Sky Time-lapse and UCF-101 datasets (4 frames skipped).

# C    Experimental Results on Long Video Generation Tasks

We obtain new state-of-the-art results on the SkyTimelapse and UCF-101 datasets for long video generation tasks. All experiments are conducted without conditional inputs. The quantitative results

are reported in Table 6. MoCoGAN, MoCoGAN-HD, DIGAN, and StyleGAN-V are GAN-based methods, which dominate the field of vision generation until 2022. Based on diffusion probabilistic models, VIDM outperforms these GAN-based methods by a large margin. However, VIDM employs the auto-regression generation strategy to generate long videos, which lacks global guidance and suffers from error accumulation. Our method, GLOBER, outperforms VIDM significantly due to its incorporation of global features and non-autoregression generation strategy. We present several video samples in Fig. 5, which demonstrate that our method can generate long videos of remarkable quality.

Table 6: Quantitative Results of FVD comparison on the SkyTimelapse and UCF-101 datasets for 128-frame long video generation.

| Method | UCF-101 | Sky Time-lapse |
|---|---|---|
| MoCoGAN [CVPR18] | 3679.0 | 575.9 |
| +StyleGAN2 backbone | 2311.3 | 272.8 |
| MoCoGAN-HD [ICLR21] | 2606.5 | 878.1 |
| DIGAN [ICLR22] | 2293.7 | 196.7 |
| StyleGAN-V [CVPR22] | 1773.4 | 197.0 |
| VIDM [AAAI23] | 1531.9 | 140.9 |
| GLOBER (ours) | **1177.4** | **125.5** |

## D  More Qualitative Results

We present more qualitative results on the UCF-101, Sky Time-lapse, and TaiChi-HD datasets in the link: `https://iva-mzsun.github.io/GLOBER`.

## E  Sensitivity Analysis of Unconditional Guidance Scale

We investigate the effectiveness of the unconditional guidance scale $\mu$ that is used when employing class-condition constraints. Table 7 presents the influence of $\mu$ on the FVD score of videos conditionally decoded by the video decoder on the UCF-101 $256^2$ benchmark. Table 8 presents the influence of $\mu$ on the FVD score of videos conditionally sampled by the video generator on the UCF-101 $256^2$ dataset. It is evident that appropriate selection of the unconditional guidance scale is important in ensuring the quality of videos decoded or sampled with class conditions.

Table 7: Sensitivity analysis of the unconditional guidance scale $\mu$ for video reconstruction on the UCF-101 dataset.

| $\mu$ | 0 | 3 | 6 | 9 | 12 | 15 |
|---|---|---|---|---|---|---|
| FVD | 211.4 | 114.3 | **106.7** | 133.2 | 281.1 | 670.8 |

Table 8: Sensitivity analysis of the unconditional guidance scale $\mu$ for video generation on the UCF-101 dataset.

| $\mu$ | 0 | 3 | 6 | 9 | 12 | 15 |
|---|---|---|---|---|---|---|
| FVD | 575.6 | 173.0 | 172.6 | 171.5 | **168.9** | 224.3 |

## F  Quality of frame features generated by Equ. (9)

When adopting the CRA loss, we directly obtain the synthesized frame feature $\bar{z}_i$ according to Equ. (9). Considering that diffusion models typically perform T denoising steps to synthesize a realistic sample, we explore the rationality of this design choice. As depicted in Fig. 6, the quality deterioration in estimated video frames is acceptable for most samples. It is reasonable since the diffusion process is under the guidance of global features, which contain sufficient local and global information.

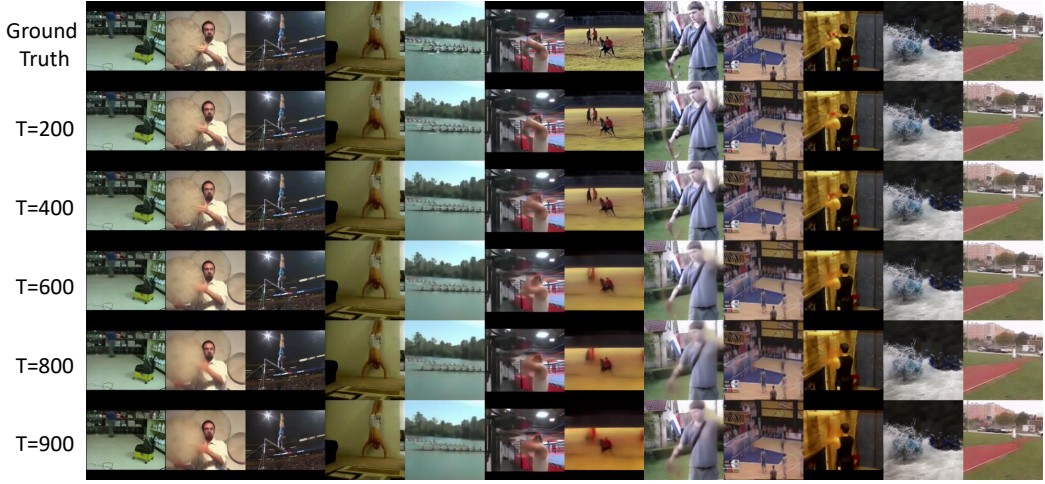

Figure 6: Video frames estimated by Equ. (9) with different timesteps on the UCF-101 dataset with $256 \times 256$ resolution.

## G   Necessity to consider "keyframes" for extracting global features from a given video

It's time- and computation- consuming to process all video frames for extracting global features, especially when we encode long videos such as 128. As depicted in Fig. 7, when the number of input video frames increasing, the maximum training batchsize dramaticly decreases and the required training time boosts. Thus it is necessary to use keyframes rather than all video frames to obtain better training efficiency.

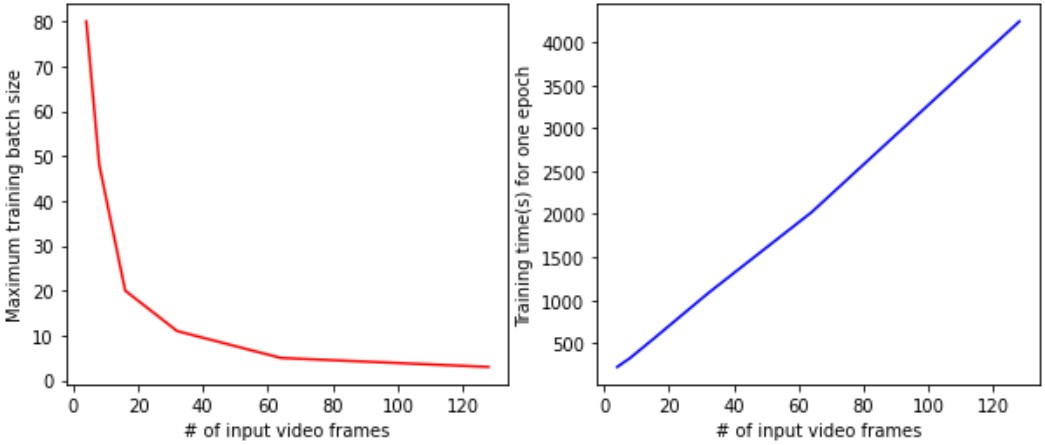

Figure 7: Comparison of the maximum training batch size and required time using different number of input video frames.

## H   More Efficiency Comparison

We compare the generation efficiency of our GLOBER with prior works, e.g. PVDM, with comparable settings as reported in Table. 9. Results with * are taken from PVDM and measured with a single NVIDIA 3090ti 24GB GPU. The rest are evaluated on a single NVIDIA 3090 24GB GPU by us due to lack of 3090ti. LVDM has not released models and scripts for 128-frame video generation. Limited by memory, our GLOBER decodes a 128-frame video by parallelly and no-overlapped

decoding every 32 video frames. DiT is the video generator of GLOBER. The results demonstrate that our GLOBER is more efficient than PVDM and LVDM when training for video generation and synthesizing long videos.

Table 9: Maximum training batch size and required inference time (in seconds) for different methods to synthesize a 256x256 resolution video.

| Method | Train Batch Size | Inference Time (16 frames) | Interence Time (128 frames) |
|---|---|---|---|
| TATS* | - | 84.8 | 434 |
| VideoGPT* | - | 139 | N/A |
| VDM* | - | 113 | N/A |
| LVDM | - | 98 | N/A |
| PVDM-L* | 2 | 20.4 | 166 |
| GLOBER (ours) | 4 | 21.4 | 145.7 |
| GLOBER (DiT only) | 8 | **3.57** | **3.57** |

## I Ablation Study on the KL loss

Following [8], the KL loss, i.e. Equ. (2), can punish the distribution of latent features towards a standard normal distribution to avoid high shift and high variance. We conduct an ablation study on the variance prediction and the KL loss. As reported in Table 10, removing variance prediction brings improvements on rFVD, but deteriorates FVD significantly since the video decoder is no longer robust to disturbance of generated features. Adding variance prediction improves the generation performance to some extent. The performance further boosts after employing the KL loss since the KL loss brings neglectable decrease on rFVD but can effectively facilitate DiT to model the distribution of global features using the diffusion theory.

Table 10: Ablation study on variance prediction and the KL loss. All experiments are conducted on the TaiChiHD dataset for 16-frame video generation with $256^2$ resolution. Autoencoders are trained for 1500 epochs (15h) and DiTs are trained for 2000 epochs (14h).

| Design of the video auto-encoder | rFVD | FVD |
|---|---|---|
| Determinstic (w/o variance) | 68.9 | 773.4 |
| + variance prediction (w/o KL loss) | 71.5 | 549.5 |
| + variance prediction + KL loss | 75.3 | 332.7 |

## J Settings of Hyper Parameters

The detailed settings of model hyper parameters are presented in Table 11.

Table 11: Hyper-parameters of the video auto-encoder and the quantitative results on video reconstruction. Experimental settings on the UCF-101 dataset are the same for both conditional and unconditional video generation except given video descriptions.

| | UCF-101 $256^2$ | Sky Time-lapse $256^2$ | TaiChi-HD $128^2$ |
|---|---|---|---|
| Batch Size | 40 | 32 | 32 |
| Learning Rate | 1e-5 | 1e-4 | 5e-5 |
| | **KL-VAE** | | |
| $f_{frame}$ | 8 | 8 | 4 |
| | **Video Encoder** | | |
| $f_{video}$ | | 2 | |
| Input Shape | | 32 | |
| Input Channels | | 4 | |
| Output Channels | | 16 | |
| Model Channels | | 320 | |
| Num Res. Blocks | | 2 | |
| Num Head Channels | | 64 | |
| Attention Resolutions | | [16, 8] | |
| Channel Multiplies | | [1, 2] | |
| | **Video Decoder (UNet)** | | |
| Input Shape | | 32 | |
| Input Channels | | 4 | |
| Output Channels | | 4 | |
| Model Channels | | 320 | |
| Num Res. Blocks | | 2 | |
| Num Head | | 8 | |
| Attention Resolutions | | [32, 16, 8] | |
| Channel Multiplies | | [1, 2, 4, 4] | |
| | **Video Generator (DiT)** | | |
| Input Shape | 16 | 16 | 16 |
| Input Channels | 16 | 16 | 16 |
| Model Channels | 1152 | 1024 | 1024 |
| Num Head | 16 | 16 | 16 |
| Depth | 28 | 20 | 20 |
| Mlp Ratio | 4 | 4 | 4 |

