# OpenReview forum: "GLOBER: Coherent Non-autoregressive Video Generation via GLOBal Guided Video DecodER"
_NeurIPS.cc/2023/Conference — NeurIPS 2023 poster_

### Official Review · Reviewer_P49b · 2023-07-07

**Soundness:** 2 fair
**Presentation:** 1 poor
**Contribution:** 2 fair
**Rating:** 4
**Confidence:** 5

**Summary:**

This paper proposes a new video generation framework based on extracting the global features of the video and conditional diffusion model to predict frame features, leading to the frame. The paper argues the proposed method outperforms prior video generation methods on various benchmarks, including UCF-101, Taichi-HD, and SkyTimelapse.

**Strengths:**

- Compared with the prior video generation methods, the proposed method considers non-autoregressive approach for generating the video, which can improve the efficiency in inference time.
- The proposed method shows better performance compared with prior works.

**Weaknesses:**

- The overall framework is quite complex, including so many notations, and a bit difficult to follow. For instance, Why $I_j$ is put into the video decoder model as well as $I_j$ in Figure 2? Moreover, are KL-VAE and video encode/decoder, discriminator jointly trained or not? What is the intuition of letting the video decoder network as a conditional diffusion model instead of letting simple 2D CNNs? Why do we need to consider "keyframes" for extracting global features from a given video? Does DiT for modeling global features is trained in post-hoc manner after the training of the entire framework?
- The paper misses an efficiency comparison with recent latent video diffusion models to improve the efficiency in training and efficiency: e.g., LVDM [He et al., 2023] and PVDM. Compared with these frameworks, what is the advantage and disadvantages of the method?
- Typo: Specificcally -> Specifically in L197.

---
[He et al., 2023] Latent Video Diffusion Models for High-Fidelity Long Video Generation
[Yu et al., 2023] Video Probabilistic Diffusion Models in Projected Latent Space, CVPR 2023

**Questions:**

- I guess the proposed method may show a worse performance if the targeting video length becomes large because the quality of global features might have a limitation and the decoder that synthesizes a frame in frame-index conditioned manner has a limited capacity. What is the (empirical) maximum length for high-quality modeling with this framework?

**Limitations:**

The paper adequately addresses the limitations in Conclusion section.

---

> ### Author Rebuttal · Authors · 2023-08-09
>
> ### R4.1 To facilitate following our framework.
> [Author] To faciliate following, we will release our codes and checkpoints as stated in the footnote of the paper.
>
> #### R4.1.1 Why Ij is put into the video decoder model as well as Ii in Figure 2?
> [Author] In Figure 2, we depict the training procedure of our framework. As specified in L164-166 and L186-188, the video discriminator takes paired frames <xi,xj> as inputs during training. Thus Ij should be inputted into the video decoder as well as Ii to obtain corresponding frames xi and xj.
>
> #### R4.1.2 Are KL-VAE and video encode/decoder, discriminator jointly trained or not?
> [Author] They are not joint trained. As specified in L102-104, KL-VAE is pretrained and fixed, and the video encoder/decoder as well as the video discriminator are trained jointly.
>
> #### R4.1.3 What is the intuition of letting the video decoder network as a conditional diffusion model instead of letting simple 2D CNNs?
> [Author] Our video auto-decoder utilizes much fewer latents to encode an input video compared to previous methods as presented in the Table Q1.3 (in **[common question 1]**). To this end, we have to rely on the powerful generation capability of the conditional diffusion model to help reconstruct local characteristics of video frames under the guidance of video global features. Moreover, training consumption can be reduced by initializing the conditional diffusion model with parameters of successful image diffusion model as specified in L49-50.
>
> #### R4.1.4 Necessity to consider "keyframes" for extracting global features from a given video.
> [Author] It’s time- and computation- consuming to process all video frames for extracting global features, especially when we encode long videos such as 128. As depicted in Fig. R4.1.4 **[in the PDF file]**, when the number of input video frames increasing, the maximum training batchsize dramaticly decreases and the required training time boosts. Thus it is necessary to use keyframes rather than all video frames to obtain better training efficiency.
>
> #### R4.1.5 Does DiT is trained in post-hoc manner?
> [Author] Yes, as specified in L191-194, DiT is separately trained to fit the distribution of global features for video generation.
>
> ### R4.2 Comparison with LVDM and PVDM.
> We find that PVDM has not provided generation codes and model checkpoints, and LVDM only released checkpoints and scripts for short-video generation. Thus, we follow the experimental setups in PVDM to compare the effeciency among PVDM, LVDM, and our GLOBER. The results are reported in Table Q1.2 (in [common question 1]). We discuss the advantages and disadvantages of our method as follows:
>
> **Advantages:**
>
> - GLOBER can take advantage of the powerful generative capability of pretrained image diffusion models (e.g. stable diffusion) to synthesize reconstructed video frames, thus requiring a much smaller dimension of latent features to represent videos as demonstrated in Table Q1.3 (in [common question 1]).
> - GLOBER is more flexible than PVDM and LVDM when decoding video frames from video latents. The video decoder in GLOBER can decode arbitrary video frames without length or interval limitations by taking the normalized indexes of target video frames as inputs.
> - GLOBER is more efficient than PVDM and LVDM when training for video generation and synthesizing long videos as demonstrated in Table Q1.2 (in [common question 1]).
> - As reported in Table. R4.2.1, GLOBER obtains better performance than PVDM and LVDM on UCF-101 for 16-frame video generation and on SkyTimelapse for 128-frame video generation.
>
> Table 4.2.1 Quantitative comparison for video generation with the resolution of 256^2. N/M-s for PVDM names N DDIM steps for genration the initial video clip and M DDIM steps for synthesing following video clips. N/M-s for GLOBER means N DDIM steps for the generation of global features and M DDIM steps for decoding video frames.
> |||UCF-101||SkyTimelapse|
> |:-:|:-:|:-:|:-:|:-:|
> || FVD16 | Total Sampling Steps | FVD128 | Total Sampling Steps |
> |StyleGAN-V | 1431.0 | - | 197.0 | - |
> |LVDM | 372 | - | 185.0 | - |
> |PVDM-S; 100/20-s | 457.4 | 100 | 159.9 | 240 |
> |PVDM-L; 200/200-s | 398.9 | 200 | 137.2 | 1600 |
> |PVDM-L;400/400-s|343.6|400|125.2|3200|
> |GLOBER(ours); 50/50-s|252.7|100|125.5|100|
> |GLOBER(ours);100/100-s|**248.9**|200|**122.4**|200|
>
> **Disadvantages of GLOBER:**
>
> - PVDM and LVDM poses stronger constraints on the correlations of adjacent video frames than GLOBER, and thus obtains better performance than GLOBER on short-video generation on simple domain dataset like SkyTimelapse and long-video generation on multi-motion datasets like UCF-101.
>
> ### R4.3 Spelling error.
> [Author] Thanks, we will fix the spelling error in the revision.
>
> ### R4.4 Explore the capacity of global features and the video decoder.
> [Author] To explore the empirical maximum length of our method, we first calculate the distribution of video length in the SkyTimelapse and TaiChiHD datasets. As depicted in Fig. R4.4.1(a) **[in the PDF file]**, TaiChiHD contains much more long videos than SkyTimelapse, thus we conduct analysis experiments on the TaiChiHD dataset. We train our video auto-encoder to model L-frame videos with (EXP1) L=16, FPS=16, 1s; (EXP2) L=64, FPS=16, 4s; (EXP3) L=128, FPS=16, 8s; (EXP4) L=256, FPS=16, 16s; (EXP5) L=1024, FPS=32, 32s. Particularly, EXPi loads the last checkpoint of EXPi-1 and is trained for 1500 epochs (15h). As depicted in Fig. R4.4.1(b), the performances of different models are comparable when the video length is no more than 512, while significantly performance drop can be seen when the video length being 1024 due to the increasement of video information. Moreover, we find that frame interpolation (FI) can help our model obtain longer videos with comparable FVD scores. In conclusion, our global features can well-capture a video with 16 seconds, and our video decoder can decode videos with FPS being at least 32.

---

> > ### Comment · Reviewer_P49b · 2023-08-17
> > **Response**
> >
> > Thanks for the response. It helps me a lot to understand the details of the method. However, at the current status, it is difficult for me to recommend acceptance. Specifically, I still have a doubt about the capability of your video autoencoder to encode possibly long videos through relatively low-dimensional latents (4,096). The response states that the proposed method can compress video (128, 256, 256, 3) to 4,096, but I really don't think this can achieve high-quality reconstructions, especially on complex datasets (e.g., UCF-101, Kinetics, and so on). In the authors' response to my review, the authors do not provide any reconstruction/generation results on such a long video with complex datasets. Without this result, it is hard for me to believe whether this method indeed scales up well to large-scale and complex datasets. In addition, the authors state "PVDM and LVDM obtain better performance than GLOBER on short-video generation on simple domain datasets like SkyTimelapse and long-video generation on multi-motion datasets like UCF-101.", I think short-video generation on simple domain dataset and long video generation on multi-motion datasets has no similarity and thus think the analysis provided in the response is not that insightful.
> >
> > Considering all of these aspects, I will retain my score.

---

> > > ### Author Response · Authors · 2023-08-17
> > > **Response to Reviewer P49b**
> > >
> > > Thank you for your response!
> > > In response to your two questions, we have the following explanations：
> > >
> > > ### R4.5: Generation results on long videos with complex datasets
> > > **We have provided the qualitative and quantitative results for 128-frame video genration on the UCF-101 dataset in the A.2 section (L9-18) in the appendix.**
> > > As reported in the Table 1 of the appendix (which is copied in the following), our GLOBER outperforms previous methods StyleGAN-V (CVPR2022) and VIDM  (AAAI2023) by a large margin.
> > > We also visualize the generated long videos on the UCF-101 and SkyTimelapse datasts in the link provided in the L21 of the appendix.
> > > Moreover, for short video generation, our method can obtain comparable performance with current SOTA models on the much more complex dataset WebVid-10M as reported in the Table Q1.4 of the common question 1.
> > >
> > > Table 1: Quantitative Results of FVD comparison on the SkyTimelapse and UCF-101 datasets for
> > > 128-frame long video generation.
> > > | Method | UCF-101 | Sky Timelapse |
> > > |:-|:-:|:-:|
> > > | MoCoGAN [CVPR18] | 3679.0 | 575.9 |
> > > | +StyleGAN2 backbone | 2311.3 | 272.8 |
> > > | MoCoGAN-HD [ICLR21] | 2606.5 | 878.1 |
> > > | DIGAN [ICLR22] | 2293.7 | 196.7 |
> > > | StyleGAN-V [CVPR22] | 1773.4 | 197.0 |
> > > | VIDM [AAAI23] | 1531.9 | 140.9 |
> > > | GLOBER (ours) | **1177.4** | **125.5**|
> > >
> > > ### R4.6: Explanation of model performance compared to PVDM and LVDM
> > > **Performance on the multi-motion dataset**
> > > Since PVDM and LVDM auto-encode a fixed number of video frames while our GLOBER pursue flexible decoding and use much less number of latent elements, PVDM and LVDM can put stronger constraint on the consistency of decoded video frames than our GLOBER when video motion is dramatic, e.g. long videos in a multi-motion dataset.
> > > Thus they obtain better performance on 128-frame video generation on UCF-101.
> > > However, for short video generation on the multi-motion dataset, our GLOBER can obtain comparable video consistency and much better video realism since PVDM and LVDM requires much more number of elements to represent a video clip than GLOBER (Table. Q1.3 in the common question 1), making their video generators difficult to fit the distribution of video latents, and our GLOBER employs the powerful diffusion model as the video decoder.
> > >
> > > **Performance on the simple domain dataset**
> > > The key reason of PVDM outperforming our GLOBER for short-video generation on simple domain dataset like SkyTimelapse lies in that since videos in such dataset contain mostly simple and statistic scenes  (city or nature scenes), the determinstic video decoder in PVDM may obtain better video reconstruction than our diffusion video decoder.
> > > Notably, despite that LVDM also employs a determinstic video decoder, it performs inferior to our GLOBER (95.2 vs 78.1 FVD) since it requires three times the number of latent features of this method, thus being too difficult to fit the latent distribution well.
> > > When the length of video increases, the video consistency of our GLOBER is still comparable with them due to videos containing small motions (clouds floating and other variation of sky), while the video realism of PVDM and LVDM drops due to error accumulation (both of them employ the auto-regressive generation strategy).
> > > Thus our GLOBER can obtain a better score against PVDM and LVDM in such case.

---

> > > ### Author Response · Authors · 2023-08-21
> > > **For Reviewer P49b**
> > >
> > > Dear Reviewer P49b,
> > >
> > > There is not so much time left for the discussion stage, if you still have questions about our work, please let us know and we will reply as soon as possible, thanks for your effort and time!
> > >
> > > Best wishes,
> > >
> > > Author

---

> > > > ### Comment · Reviewer_P49b · 2023-08-21
> > > > **Response**
> > > >
> > > > Thanks for the detailed follow-up. For me, the limitation of GLOBER on long-video, multi-motion setup is quite critical issue. Specifically, it seems the scalability of the method toward real-world complex and long videos is limited, in contrast to other methods; so forcing the latent dimension to be small (4,096) is not persuable. If scalability can be ensured with low-dimensional latents for video encoding, it is definitely a good contribution and a good method, but if not, I think the strength of the method is limited. Therefore it is a bit difficult to recommend acceptance. I think in the future, the authors can provide long vid. experiments using multi-motion datasets with higher-dimensional latents (like 16,384) and check whether such change can outperform previous arts such as PVDM to improve the soundness of the method, not just stating the method shows limited performance due to the usage of low-dimensional latents.

---

> > > > > ### Author Response · Authors · 2023-08-21
> > > > > **Response to Reviewer P49b**
> > > > >
> > > > > Thanks for your response!
> > > > >
> > > > > ### R4.7:
> > > > > Firstly, we did not force the number of latent elements to be small (4096).
> > > > > The focus is that dispite of small number of latent elements, our proposed method can still obtain SOTA results on multiple multi-motion benchmarks for short video generation, i.e. UCF-101 and Webvid-10M, and outperform most contemporary methods like VIDM for long video generation.
> > > > >
> > > > > Secondly, since we employ much less number of latent elements than PVDM (4096 vs 65536) to represent a long video, we are afraid that harse comparison with PVDM is unfair for our proposed method, especially when we have outperform PVDM for 16-frame video genration on the multi-motion dataset UCF-101 by ~100 FVD and obtain other advantages like flexible decoding and efficiency.
> > > > > We will add the experiment you mentioned on multi-motion datasets with higher-dimensional latents in our paper later.
> > > > >
> > > > > Thanks again for your effort and time to serve as a reviewer!
> > > > >
> > > > > Best regrads,
> > > > > Author

---

### Official Review · Reviewer_JPZt · 2023-07-07

**Soundness:** 3 good
**Presentation:** 2 fair
**Contribution:** 4 excellent
**Rating:** 6
**Confidence:** 4

**Summary:**

This work introduces a novel non-autoregressive method, GLOBER, that first generates global features for comprehensive global guidance and then synthesizes video frames based on these global features to produce coherent videos. The authors propose a video auto-encoder to encode videos into global features and a video decoder to decode the global features and synthesize video frames in a non-autoregressive manner. Notably, the video decoder uses normalized frame indexes to perceive temporal information, allowing it to synthesize any video clips with predetermined frame indexes. The authors also introduce a unique adversarial loss to enhance global coherence and local realism of the synthesized video frames. Finally, a diffusion-based video generator is employed to fit the global features produced by the video encoder for video generation. The effectiveness and efficiency of the proposed method are demonstrated through extensive experiments, and it sets new state-of-the-art results on multiple benchmarks.

**Strengths:**

1. The inclusion of Coherence and Realism Adversarial Loss is a novel approach compared to previous diffusion-based architectures.
2. Extensive experiments have been performed on various benchmarks, all demonstrating the significance of GLOBER.

**Weaknesses:**

1. The authors identify VideoFusion as the most closely related work due to its use of non-autoregressive generation. However, there are other public models, such as ModelScope Text-to-Video, that use non-autoregressive generation in the latent space similarly to the authors' work. I suggest that the authors compare their work to these models as well.
2. It appears that GLOBER outperforms VideoFusion in all tasks, which generates videos in the pixel space. This superiority seems to result from the CRA loss proposed by the authors. Therefore, a direct comparison between GLOBER (without CRA loss) and VideoFusion would be intuitive. However, inconsistent results in Table 1 and 3 make this comparison unfeasible. Could the authors explain this inconsistency and provide justifications for GLOBER's superiority over VideoFusion, aside from the CRA loss?
3. My interpretation of Equation 9 suggests it's an estimation of the frame feature. However, this estimation might not be accurate because, like in DDPM (or DDIM), one could perform T steps of reverse denoising to generate images. What is the quality difference between these two types of images? I assume that the frame feature generated by Equation 9 will be of lower quality.
4. In Table 2, why does GLOBER use 50+50 diffusion steps?
5. The optimization objective of Equation 2 in the Video Encoding section appears to be derived from the Variational Autoencoder. Could the authors provide a justification for this design? My understanding is that video encoding trains a dataset-dependent distribution to be sampled as z_t.
6. In Line 127, "Gauss distribution" appears to be misspelled.
7. The authors seem to have overlooked specifying the dimension of C'.

My score could be revised upward if my concerns are adequately addressed.

**Reference:**

[1] ModelScope Text-to-Video Technical Report, arXiv. (Model: **https://modelscope.cn/models/damo/text-to-video-synthesis/summary**).

**Questions:**

Please refer to the weaknesses.

**Limitations:**

In this work, the authors mentioned limitations and broader impact.

---

> ### Author Rebuttal · Authors · 2023-08-08
>
> ### R3.1 More comparison with contemporary methods.
> We present more efficiency and quality comparsion with contemporary methods like Modelscope, LVDM, and PVDM in our global rebuttal **[Common Question 1]**, please view it for more details.
>
> ### R3.2
> #### R3.2.1 Inconsistency of Table 1 and Table 3.
> [Author] Table 1 reports the generation performance that involves both the video auto-encoder and DiT, while Table 3 measures the reconstruction FVD that only involves the video auto-encoder as specified in L276-277. We will modify the FVD in Table 3 to FVDrec to avoid confusion in the paper later.
> #### R3.2.2 Comparison of VideoFusion and GLOBER without CRA is unfair.
> [Author] Different from VideoFusion, which naturally incorporates constraints on the local consistency of frame-wise characteristics by decomposing video components, our GLOBER requires the CRA loss to obtain local constraints by punishing video frames that violate such consistency. Thus it may be unfair for our GLOBER to compare with VideoFusion without the CRA loss.
>
> ### R3.3 Quality difference between two types of images is acceptable.
> [Author] As depicted in Fig. R3.3.1 **[in the PDF file]**, the quality deterioration in estimated video frames is acceptable for most samples. It is reasonable since the diffusion process is under the guidance of global features, which contain sufficient local and global information.
>
> ### R3.4 Why we use 50+50 diffusion steps in GLOBER.
> [Author] As specified in L220-224, 50+50 diffusion steps are the default settings for all our experiments. Such setups follow the setting of stable diffusion, which uses 50 ddim steps in default, and obtains well efficiency and generation quality as reported in our experiments.
>
> ### R3.5 The design of Equation 2.
> [Author] Our target is exactly to train global features to conform to a dataset-dependent distribution similar to z_t. In fact, the distribution of z_t is also punished towards a standard normal distribution during training by employing the KL loss (Equation 2) to avoid high shift and high variance, as specified in [1]. The formulation of the KL loss is derived from the KL divergence between two multi-variable Gauss distributions (one is the distribution outputted by the video encoder and another is the standard Gauss distribution), and the KL-penalty is slight with a small loss weight of 1e-6 as specified in L217 following [1], thus the final distribution of global features is still dataset-dependent. The effectiveness of the KL loss has been proven in [1]. For better understanding, we conduct an ablation study on the variance prediction and the KL loss. As reported in Table R3.5.1, removing variance prediction brings improvements on FVDrec, but deteriorates FVDgen significantly since the video decoder is no longer robust to disturbance of generated features. Adding variance prediction improves the generation performance to some extent. The performance further boosts after employing the KL loss since the KL loss brings neglectable decrease on FVDrec but can effectively facilitate DiT to model the distribution of global features using the diffusion theory.
>
> Table R3.5.1 Ablation study on variance prediction and the KL loss. All experiments are conducted on the TaiChiHD dataset for 16-frame video generation with 256^2 resolution. Autoencoders are trained for 1500 epochs (15h) and DiTs are trained for 2000 epochs (14h).
> | Design of the video auto-encoder | FVDrec | FVDgen |
> | :-: | :-: | :-: |
> | Determinstic (w/o variance) | 68.9 | 773.4 |
> | +variance prediction (w/o KL loss) | 71.5 | 549.5 |
> | +variance prediction + KL loss | 75.3 | 332.7 |
>
> [1] R. Rombach, A. Blattmann, D. Lorenz, P. Esser, and B. Ommer, “High-resolution image synthesis with latent diffusion models”.
>
> ### R3.6 Spelling Error.
> [Author] Thank you for bringing this to our attention. We will rectify the spelling error in revision.
>
> ### R3.7 The dimension of C'.
> [Author] The dimension of C’ is 4 for 256x256 resolution and 3 for 128x128 resolution. We will add explanations in the revision.

---

> > ### Comment · Reviewer_JPZt · 2023-08-19
> > **Post-rebuttal discussions**
> >
> > Dear the authors of Paper 6343,
> >
> > Many thanks for your detailed reply. It solves most of my concern. I have one follow-up question about Question3.5. Is the variance prediction in Table R3.5.1 indicates the sampling process in Eq.1? Btw, is the auto-encoding stage in [1] trained in an end-to-end manner with the denoising UNet using KL loss, like what GLOBER does?
> >
> > I look forward to your reply.
> >
> > Kind regards,
> >
> > Reviewer JPZt

---

> > > ### Author Response · Authors · 2023-08-19
> > > **Response to Reviewer JPZt**
> > >
> > > Thank you for your response!
> > >
> > > For the first question, **yes**, the variance prediction in Table R3.5.1 is the sampling process in Eq. 1.
> > >
> > > For the second question, **no**, the auto-encoding stage in [1] utilized simple CNNs as its encoder and decoder. The overall structure of the auto-encoder in [1] (i.e. KL-VAE) is similar to the traditional VQ-VAE [2] or VQ-GAN [3] except for that KL-VAE represents images with continuous latent features while VQ-VAE and VQ-GAN represents images with discrete tokens through vector-quantization, which are difinitely different from our model.
> > >
> > > [2] Zero-Shot Text-to-Image Generation.
> > >
> > > [3] Taming Transformers for High-Resolution Image Synthesize.

---

> > > > ### Comment · Reviewer_JPZt · 2023-08-19
> > > > **Post-rebuttal discussions (2)**
> > > >
> > > > Dear the authors of Paper 6343,
> > > >
> > > > Thank you for your thorough response. Your rebuttal has addressed my concerns, and as promised, I have decided to raise the score.
> > > >
> > > > Best regards,
> > > >
> > > > Reviewer JPZt

---

> > > > > ### Author Response · Authors · 2023-08-19
> > > > > **Response to Reviewer JPZt**
> > > > >
> > > > > Thank you for recognizing our work!
> > > > >
> > > > > Best wishes,
> > > > >
> > > > > Author

---

### Official Review · Reviewer_xuBL · 2023-07-07

**Soundness:** 3 good
**Presentation:** 3 good
**Contribution:** 2 fair
**Rating:** 5
**Confidence:** 5

**Summary:**

In this paper, the author studies the text-to-video task and proposes a method called GLOBER. The proposed method first generates a global guidance feature, then the video frames are generated through a diffusion model that takes the frame index as a condition. An adversarial loss is also proposed to improve global coherence and local realism.

**Strengths:**

1. The overall presentation of the proposed method is clear and easy to follow.
2. The author conducts experiments on three widely used datasets (e.g., Sky Time-lapse, TaiChi-HD and UCF-101).

**Weaknesses:**

1. As the author claims their method is capable of generating videos from text. It would be great if the author could compare their methods with SOTA open-sourced methods (DAMO-text2video, VideoCrafters, CogVideo and VideoFactory) on WebVid-10M.
2. The author should also consider comparing their methods with PVDM.


**Questions:**

Please refer to the weakness part.

**Limitations:**

Please refer to the weakness part.

---

> ### Author Rebuttal · Authors · 2023-08-09
>
> ### R2.1 Experiments on WebVid-10M.
> [Author] We present quality comparsion with ModelScope (DAMO-text2video)[1], VideoCrafters[2], LVDM[3], and VideoFactory[4] on WebVid-10M in our global rebuttal **[Common Question 1]**, please view it for more details. We find the open-sourced code of CogVideo does not support parallel generation given different input descripitons and requires ~10 minutes to synthesize a video, which is time-consuming thus we only add comparison with CogVideo on the UCF-101 dataset.
>
> [1] VideoFusion: Decomposed diffusion models for high-quality video generation. 2023. CVPR.
> [2] Inference with codes from https://github.com/VideoCrafter/VideoCrafter
> [3] Latent video diffusion models for high-fidelity long video generation. 2022. ARXIV.
> [4] VideoFactory: Swap Attention in Spatiotemporal Diffusions for Text-to-Video Generation. 2023. ARXIV.
>
> ### R2.2 Comparison with PVDM.
> Thanks for bringing this to our attention, we discuss the differences between PVDM and our GLOBER, and then analyze the advantages and disadvantages of our GLOBER compared to PVDM as follows (all reported results of GLOBER are evaluated using previous checkpoints), which we will add to the paper later.
>
> #### Differences
> - Motivation and auto-encoder: To address the computation- and memory-inefficiency problem of video generation, PVDM utilizes 3D-to-2D projection to encode a video into three 2D latent features, and reconstructs videos in a deterministic manner, where the video decoder directly decodes local characteristics of reconstructed frames from the latent features. Different from PVDM, our GLOBER focuses on providing global guidance for long-video generation and encodes a video into a single latent feature. Thus GLOBER utilizes the powerful generation capability of the pretrained image diffusion model to synthesize local characteristics of reconstructed images, which significantly reduces the total dimension of flattened latent features.
>
> - Video generator and generation strategy: The video generator of PVDM has to fit three 2D latents simultaneously for each video, and modeling long video generation in an autoregressive manner (i.e. generating following 16 frames given previous 16 generated frames). However, the video generator of GLOBER only needs to model one 2D latent for each video and employs a non-autoregressive generation strategy to synthesize long videos (i.e. generating 128 frames with one sampling).
>
> #### Advantages of GLOBER compared to PVDM:
> -	GLOBER can take advantage of the powerful generative capability of pretrained image diffusion models (e.g. stable diffusion) to synthesize local characteristics of reconstructed video frames under the guidance of global features, thus requiring a much smaller dimension of latent features to represent videos as demonstrated in Table R2.2.1.
>
> Table R2.2.1 Comparison on the dimension of flattened latent features for representing a T-frame video with HxW resolution. LVDM encodes videos using 3D CNN with the spatial and temporal downsampling rates being 8 and 4 respectively, obtaining latents with shape (T//4, H//8, W//8, 3). PVDM utilizes 3D-to-2D projection to encode a video into three 2D latents with shape (T, H//d, 4), (T, W//d, 4) and (H//d, W//d, 4) respectively, with d=8 in default. GLOBER extracts video global information and encodes an input video into a latent feature with shape (H//16, W//16, 16).
> | （T, H, W, C） |  LVDM  | PVDM | GLOBER(ours) |
> | :---: | :---: | :---: | :---: |
> | (16, 256, 256, 3) | 12288 | 8192 | **4096** |
> | (128, 256, 256, 3) | 98304 | 65536 | **4096** |
>
> - GLOBER is more flexible than PVDM when decoding video frames from video latents. In PVDM, the video decoder reconstructs video frames with fixed length (i.e. 16 frames) and fixed interval (i.e. predefined FPS). However, the video decoder in GLOBER can decode arbitrary video frames without any length or interval limitation by taking the normalized indexes of target video frames as inputs.
>
> - GLOBER is more efficient than PVDM when training for video generation and synthesizing long videos as demonstrated in Table Q1.2 in **[Common Question 1]**.
>
> - As reported in Table. R2.2.2, GLOBER obtains better performance than PVDM on UCF-101 for 16-frame video generation and on SkyTimelapse for 128-frame video generation. It is reasonable since by initializing from pretrained image diffusion model, the video decoder in GLOBER becomes much more powerful than that in PVDM, thus enhancing the generation of complex videos like UCF-101. For the small domain dataset SkyTimelapse, the global guidance provided by our GLOBER is able to improve the global coherence of synthesized long videos, thus obtaining better FVD.
>
> Table R2.2.2 Quantitative comparison for video generation with the resolution of 256x256. N/M-s for PVDM names N DDIM steps for genration the initial video clip and M DDIM steps for synthesing following video clips. N/M-s for GLOBER means N DDIM steps for the generation of global features and M DDIM steps for decoding video frames.
> | |  |  UCF-101 |  | SkyTimelapse |
> | :---: | :---: | :---: | :---: | :---: |
> | | FVD16 | Total Sampling Steps &#124; | FVD128 | Total Sampling Steps |
> | StyleGAN-V | 1431.0 | - | 197.0 | - |
> | PVDM-S; 100/20-s | 457.4 | 100 | 159.9 | 240 |
> | PVDM-L; 200/200-s | 398.9 | 200 | 137.2 | 1600 |
> | PVDM-L; 400/400-s | 343.6 | 400 | 125.2 | 3200 |
> | GLOBER (ours); 50/50-s | 252.7 | 100 | 125.5 | 100 |
> | GLOBER (ours); 100/100-s| **248.9** | 200 | **122.4** | 200 |
>
> #### Disadvantages of GLOBER compared to PVDM:
> - PVDM poses stronger constraints on the correlations of adjacent video frames than GLOBER, thus obtaining better performance than GLOBER in short-video generation on simple domain dataset like SkyTimelapse and long-video generation on multi-action datasets like UCF-101.

---

> > ### Comment · Reviewer_xuBL · 2023-08-18
> >
> > Thanks for the rebuttal. I think most of my concerns have been well addressed. I will raise my score.

---

> > > ### Author Response · Authors · 2023-08-18
> > > **Response to Reviewer xuBL**
> > >
> > > Thank you for your recognition of our job!  We will incorporate the above discussions when we revise our paper.

---

### Official Review · Reviewer_4poX · 2023-07-08

**Soundness:** 3 good
**Presentation:** 3 good
**Contribution:** 3 good
**Rating:** 6
**Confidence:** 4

**Summary:**

The study introduces a unique non-autoregressive approach called GLOBER. This method initially generates global features, offering a thorough global guidance, which then synthesizes video frames using these global features to produce cohesive videos. Furthermore, the study suggests a coherence and realism adversarial loss to improve the quality of the videos.

**Strengths:**

The suggested non-autoregressive technique is simple and effective.

The empirical tests and ablation studies conducted are adequate

**Weaknesses:**

For Table 2, the absence of some contemporary methods such as modelscope implies that the assertion regarding inference time and GPU memory may not be as robust as claimed.

What strategies are in place to ensure that the distribution of global features produced by DiT during inference aligns with the features acquired by the video encoder during the training phase?

I'm intrigued to find out whether this non-autoregressive approach is effective with lengthy videos, for example, those consisting of 128 or 256 frames. If it's not, the benefits of this method could be significantly reduced.

**Questions:**

Please refer weakness

**Limitations:**

The authors have discussed limitations.

---

> ### Author Rebuttal · Authors · 2023-08-08
>
> ### R1.1 More efficiency comparison with contemporary methods.
> [Author] We present more efficiency and quality comparsion with contemporary methods like Modelscope, LVDM, and PVDM in our global rebuttal **[Common Question 1]**, please view it for more details.
>
>
> ### R1.2 Align the distribution of global features.
> [Author] We first train the video auto-encoder as well as the discriminator jointly until convergence, and then train DiT with the parameters of the video auto-encoder being fixed to ensure the alignment. As specified in L53-54 and L191-194, DiT is an independent generative model and is optimized after we finish training the video auto-encoder. The training setups for the video auto-encoder and DiT are also separately presented in Section 4.1.
>
> ### R1.3 The effectiveness of GLOBER on synthesizing long videos.
> [Author] Given that 128 is a commonly used length when testing long video generation, we have presented quantitative and qualitative results of 128-frame video generation on the UCF-101 and Sky Time-lapse datasets with 256x256 resolution in the appendix, which demonstrate the effectiveness of our GLOBER on long video generation. Samples are also visualized in the link of the A.3 section in the appendix.

---

> > ### Comment · Reviewer_4poX · 2023-08-19
> >
> > Thanks for your responses. My concerns have been well addressed.  But as a video generation paper, it is necessary to provide the video results,  in addition to quantitative results. Could you please provide an anonymous links containing video results? As indicated in PC emails, such a link is allowed and encouraged for video generation papers.

---

> > > ### Author Response · Authors · 2023-08-19
> > > **Response to Reviewer 4poX**
> > >
> > > We have provide an anonymous link in Section A.3 (L20-21) of the appendix, and that is https://anonymouss765.github.io/GLOBER.
> > > Please refer to this link for both short and long video samples!
> > >
> > > Thank you for your response!

---

> > > > ### Comment · Reviewer_4poX · 2023-08-19
> > > >
> > > > Thanks! I raise my score and hope this work can be open-sourced, which can significantly benefit  community.

---

> > > > > ### Author Response · Authors · 2023-08-19
> > > > > **Response to Reviewer 4poX**
> > > > >
> > > > > Thank you for recognization! To facilitate following researches, we will open source our codes and models once the paper is accepted.

---

### Author Rebuttal · Authors · 2023-08-09

We sincerely thank all reviewers for your valuable reviews and suggestions. We have carefully replied to each question and we are welcome for more discussions!!

### [Common Question 1] More comparison with contemporary methods.
### Efficiency comparison:
[Author] We add comparisons with contemporary methods such as modelscope and LVDM in Table Q1.1, where all settings are the same as that in Table 2. We also compare with the recent method PVDM as reported in Table Q1.2 using a 3090 GPU due to lacking 3090ti GPUs. Our method obtains outstanding efficiency for the following two reasons:
- 1) We adopt a non-autoregressive strategy to synthesize all video frames with only one sampling while most methods employ either the autoregression or interpolation strategy, which requires multiple sampling to create a long video.
- 2) We use a latent vector (i.e. global feature) with the least number of elements to represent an input video as demonstrated in Table Q1.3 Notably, many methods (including VIDM, VDM, and VideoFusion) model video generation at the frame level, thus being time-consuming.

Table Q1.1 Comparison of sampling time/memory using different methods for generating multiple video frames with 256x256 resolution, 1 batch size, default diffusion steps, and comparable GPU memory on a v100 GPU. F represents the number of video frames.
|         | VIDM | VDM| LVDM | Modelscope | VideoFusion | TATS | GLOBER(ours) |
| :-------: |  :-------: |  :-------: |  :-------: |  :-------: |  :-------: |  :-------: |   :-------: |
| DDIM Steps | 100     | 100    | 50    | 50 | 50 | N/A | 50+50 |
| F=16   | 192s/20G | 125s/11G | 75s/9G | 31s/6G | 22s/7G | **6s**/16G | **6s**/7G |
| F=32   | 375s/20G | 234s/11G | 141s/13G | 48s/8G | 39s/9G | 26s/16G | **11s**/11G |
| F=64   | 771s/20G | 329s/11G | 288s/20G | 82s/12G | 76s/13G | 65s/16G | **21s**/19G |

Table Q1.2 Maximum training batch size and required inference time (in seconds) for different methods to synthesize a 256x256 resolution video. Results with * are taken from PVDM and measured with a single NVIDIA 3090ti 24GB GPU. The rest are evaluated on a single NVIDIA 3090 24GB GPU by us due to lack of 3090ti. LVDM has not released models and scripts for 128-frame video generation. Limited by memory, our GLOBER decodes a 128-frame video by parallelly and no-overlapped decoding every 32 video frames. DiT is the video generator of GLOBER.
|               | Train Batch Size | Inference Time(16-frame) |  Inference Time(128-frame) |
|  :-------: |  :-------: |  :-------: |   :-------: |
| TATS* | - | 84.8 | 434 |
| VideoGPT* | - | 139 | N/A |
| VDM* | - | 113 | N/A |
| LVDM | - | 98 | N/A|
|PVDM-L*| 2 | 20.4 | 166 |
|GLOBER(ours) | 4 | 21.4 | 145.7 |
|GLOBER(DiT only) | **8** | **3.57** | **3.57** |

Table Q1.3 Model design of different methods and comparison of the number of elements used to represent a T-frame video with HxW resolution. LVDM encodes videos using 3D CNN with the spatial and temporal downsampling rates being 8 and 4 respectively, obtaining latent features with shape (T//4, H//8, W//8, 3). PVDM utilizes 3D-to-2D projection to encode a video into three 2D latent features with shape (T, H//d, 4), (T, W//d, 4) and (H//d, W//d, 4) respectively, with d=8 in default. TATS uses 3D VQGAN to encode videos with the spatial and temporal downsampling rates being 8 and 4 respectively, obtaining discrete video latent feature with shape (T//4, H//8, W//8). Modelscope adopts KL-VAE to encode videos by frame, obtaining video latent feature with shape (T, H//16, W//16, 4). GLOBER encodes video global information into a latent feature of shape (H//16, W//16, 16), which is time-independent and can be flexibly decoded into video frames with arbitrary FPS.
|  | VIDM | VDM | LVDM | PVDM | VideoFusion | TATS | Modelscope | GLOBER (ours) |
|  :--: |  :--: |  :--: |   :--: |  :--: |  :--: |  :--: |   :--: |    :--: |
| Non-arutoregression | &#10008; | &#10008; | &#10008; | &#10008; | &#10004; | &#10008; | &#10004; | &#10004; |
| Video Encoding         |  &#10008; | &#10008; | &#10004; | &#10004; | &#10008; | &#10004; | &#10004; | &#10004; |
| (16, 256, 256, 3)       | - | - | 12288 | 8192 | - | **4096** | 16384 | **4096** |
| (128, 256, 256, 3)     | - | - | 98304 | 65536 | - | 32768 | 131072 | **4096** |

### Quality Comparison
We add comparisons with contemporary methods on UCF-101 in Table Q1.4 and train our method on the webvid-10M dataset for 5 epochs to compare with other SOTA methods as reported in Table Q1.5. Our GLOBER significantly outperforms other methods on the UCF-101 dataset and obtains comparable performance on the Webvid-10M dataset. It can be attributed to two main reasons:
- 1. By initializing our video decoder with pretrained Stable Diffusion, we inherit its powerful ability to synthesize high-quality video frames.
- 2. The reduction in the element number of video latents makes it easier for our video generator to fit the distribution of video latents for the video generation task.

Table Q1.4 Quantitative comparison for video generation with the resolution of 256x256 on UCF-101.
| Method | Zero-shot | FVD |
| :---: | :---: | :---: |
| CogVideo | &#10004; | 701.6 |
| MagicVideo|  &#10004; | 699.0 |
| LVDM | &#10004; | 641.8 |
| ModelScope | &#10004; | 639.9 |
| Video LDM | &#10004; | 550.6 |
| VideoCrafters | &#10004; | 516.2 |
| VideoFactory | &#10004; | 410.0 |
| VideoGPT | &#10008; | 2880.6 |
| MoCoGAN | &#10008; | 2886.8 |
| StyleGAN-V | &#10008; | 1431.0 |
| CogVideo | &#10008; | 626 |
| LVDM | &#10008; | 372 |
| PVDM | &#10008; | 343.6 |
| GLOBER (ours) | &#10008; | **252.7** |

Table Q1.5 Quantitative comparison for video generation with the resolution of 256x256 on Webvid-10M.
| Method | FVD | CLIPSIM |
| :---: | :---: | :---: |
| VideoCrafters | 759.30 | 0.2981 |
| LVDM | 455.53 | 0.2751 |
| ModelScope | 414.11 | 0.3000 |
| VideoFactory | 292.35 | **0.3070** |
| GLOBER (ours) | **234.84** | 0.2816 |

---

### Comment · Area_Chair_316v · 2023-08-18
**Discussion period**

Dear reviewers 4poX and JPZt

You haven't provided your comments on the authors' response. Please do so before the deadline. Does the response address any of your concerns?

The AC thanks to reviewers xuBL and P49b for doing so!

AC

---

### Decision · Program_Chairs · 2023-09-21

**Decision:**

Accept (poster)

**Comment:**

The paper received in-depth reviews, in which they highlighted the overall presentations, extensive experiments, results superior to prior baselines and improved inference time. The latter is particularly interesting as recent methods tend to require a lot of compute to generate content. At the same time they also raised a lot of questions. During the discussion period the authors did a good job in addressing most of the concerns leading to increased scores overall. Reviewer P49b provided additional concerns and remained not convinced about the paper. While their concerns have merit, the AC believes that the manuscript is an interesting piece of work and, hence, recommends acceptance.